# Bisphenol S Reduces Pig Spermatozoa Motility through Different Intracellular Pathways and Mechanisms than Its Analog Bisphenol A

**DOI:** 10.3390/ijms24119598

**Published:** 2023-05-31

**Authors:** Mercedes Torres-Badia, David Martin-Hidalgo, Rebeca Serrano, Luis J. Garcia-Marin, Maria J. Bragado

**Affiliations:** 1Research Group of Intracellular Signaling and Technology of Reproduction (SINTREP), Institute of Biotechnology in Agriculture and Livestock (INBIO G+C), University of Extremadura, 10003 Cáceres, Spain; mtorresbdp@alumnos.unex.es (M.T.-B.); davidmh@unex.es (D.M.-H.); rebecasp@unex.es (R.S.); ljgarcia@unex.es (L.J.G.-M.); 2Research Unit, Complejo Hospitalario Universitario de Cáceres, 10003 Cáceres, Spain

**Keywords:** bisphenols, BPS, toxicology, spermatozoa, motility, phosphorylation

## Abstract

Bisphenol A (BPA: 2,3-bis (4-hydroxyphenyl) propane) is an environmental chemical widely used in the manufacturing of epoxy polymers and many thermoplastic consumer products. Serious concerns about its safety led to the development of analogs, such as BPS (4-hydroxyphenyl sulfone). Very limited studies about BPS’s impact on reproduction, specifically in spermatozoa, exist in comparison with BPA. Therefore, this work aims to study the in vitro impact of BPS in pig spermatozoa in comparison with BPA, focusing on sperm motility, intracellular signaling pathways and functional sperm parameters. We have used porcine spermatozoa as an optimal and validated in vitro cell model to investigate sperm toxicity. Pig spermatozoa were exposed to 1 and 100 μM BPS or BPA for 3 and 20 h. Both bisphenol S and A (100 μM) significantly reduce pig sperm motility in a time-dependent manner, although BPS exerts a lower and slower effect than BPA. Moreover, BPS (100 μM, 20 h) causes a significant increase in the mitochondrial reactive species, whereas it does not affect sperm viability, mitochondrial membrane potential, cell reactive oxygen species, GSK3α/β phosphorylation or phosphorylation of PKA substrates. However, BPA (100 μM, 20 h) leads to a decrease in sperm viability, mitochondrial membrane potential, GSK3β phosphorylation and PKA phosphorylation, also causing an increase in cell reactive oxygen species and mitochondrial reactive species. These intracellular effects and signaling pathways inhibited might contribute to explaining the BPA-triggered reduction in pig sperm motility. However, the intracellular pathways and mechanisms triggered by BPS are different, and the BPS-caused reduction in motility can be only partially attributed to an increase in mitochondrial oxidant species.

## 1. Introduction

Bisphenol molecules, particularly bisphenol A (BPA: 2,2-bis (4-hydroxyphenyl) propane), are environmental chemicals widely used in the manufacturing of epoxy polymers and many thermoplastic consumer products such as food container coating, water pipes, paper receipts and electronics [1]. Therefore, animals and humans are routinely exposed to it in different ways, including a dietary pathway [1,2]. In fact, the presence of BPA has been evidenced in human breast milk, umbilical cord, placental tissue, and urine [2].

Research published on BPA’s impact on animal and human health has reported adverse effects on development as well as on cardiovascular, nervous and reproductive functions [3,4]. Due to serious concerns about the safety of BPA, along with its extensive use and exposure through different pathways, BPA utilization was prohibited in the manufacturing of nursing bottles in the European Union and Canada [5,6], which triggered the development of structural analogs of BPA such as bisphenol F (BPF: 4,4′-methylenediphenol), bisphenol B (BPB: 2,2′-bis (4-hydroxyphenyl) butane) and bisphenol S (BPS: 4-hydroxyphenyl sulfone). In fact, besides BPA, similar amounts of its analogs BPF, BPS and BPB have been found in foodstuffs and beverages in United States and Asia [7].

Research studies about these newest BPA analogs and their potential health implications showed several adverse effects including cytotoxicity, neurotoxicity, endocrine disruption and reproductive toxicity [5,8,9,10,11]. Moreover, it has been described that BPA and its analogs may act as endocrine-disrupting molecules [3,12] with estrogenic activity [13].

As mentioned above, BPS was introduced in the market as a potentially safer alternative to BPA, although, as well as another BPA analog, BPF, it is currently not regulated, and the tolerable dose intake has not been identified yet [14]. However, BPS production and use have been increasing over the past few years, leading to the detection of BPS in environmental samples (CDC 2019), thermal receipts and canned and pre-packaged foods [15,16]. In a previous paper, it was reported that BPS presents a short half-life in humans as it is absorbed and excreted within hours after exposure [17]. Interestingly, increasing amounts of BPS have been found in human urine samples over the time in United States [18] and Asia [19]. Regarding the impact of bisphenols on reproduction, BPS acts in vivo through an endocrine-disruptor mechanism that negatively affects the number of ovarian follicles and oocyte quality in rats [20] as well as testosterone levels in mice [21]. A recent study showed a negative association between BPS and human semen quality as men with detectable BPS concentrations in urine samples presented lower semen quality parameters [22].

Although studies about the negative impact of BPA in mammalian spermatozoa have been performed in rats [23,24], mice [25,26,27], bulls [28] and humans [22,29], research about the effects of BPS on male reproductive health outcomes, specifically on spermatozoa physiology, is recent and very limited [23,28,29,30,31]. On one hand, BPS did not significantly affect any human sperm parameter, whereas BPA led to 100% of spermatozoa being immotile, an 80% decrease in viability and a 90% reduction in high mitochondrial membrane potential [31]. On the other hand, in bovine sperm, treatment with both bisphenols reduced progressive motility but to different extents; while BPS reduced it by 45%, BPA decreased it by 60% [28]. However, another study in spermatozoa and oocytes from the same species has found that BPA caused higher oxidative stress than BPS or PBF, suggesting that they are likely acting through different mechanisms [28].

Therefore, the aim of this work was to study the in vitro functional effects of BPS in comparison with BPA in pig spermatozoa, focusing on sperm motility, and to further investigate sperm intracellular signaling pathways and the cell processes involved. For this purpose, we have considered the porcine spermatozoa as an optimal and well-validated in vitro cell model to investigate sperm toxicity [32,33,34,35] and also because of the great importance that pig sperm functionality has in worldwide agricultural practices due to its enormous economic impact on livestock production. In addition, porcine spermatozoa represent a well-described cell model in the sperm field due to its advantageous translation into human assisted reproduction techniques [36]. The results of this work will contribute to deepening the understanding of the impact of the BPA analog BPS, the production and use and therefore possible effects of which are increasing, on male reproductive health.

## 2. Results

### 2.1. Effects of 4,4′-Sulfonyldiphenol (BPS) and Bisphenol A (BPA) on Pig Spermatozoa Motility

To evaluate the effect of BPS in comparison with BPA on motility parameters, pig spermatozoa were incubated for 3 and 20 h in the absence or presence of 1 µM and 100 µM of BPS or BPA. As Figure 1 shows, sperm incubation with 1 µM of each bisphenol does not affect sperm motility. However, 100 µM BPS or BPA significantly decreased total sperm motility at 3 and 20 h of incubation. The intensity of this effect at 3 h incubation varies with the bisphenol used (Figure 1A), as BPA 100 µM causes a higher reduction in motility than its analog BPS (48% vs. 18% decrease, respectively) at the same concentration. After 20 h of incubation, both bisphenols, BPA and BPS 100 µM, show a significant and similar decrease in motility (Figure 1B).

Focusing on the impact of bisphenols on the progressive motility, the concentration of 1 µM does not cause any effect at any time (Figure 2), whereas 100 µM of BPA or BPS causes significant differences at 3 and 20 h treatment. As shown in Figure 2A, the BPS reduction in the percentage of progressive spermatozoa is smaller at 3 h treatment than the effect of BPA (22% vs. 60% decrease, respectively). However, 20 h treatment with bisphenols (100 µM) significantly decreased (Figure 2B) the percentage of progressive spermatozoa, leading to a similar effect between BPS and BPA (91% vs. 97% decrease, respectively).

Regarding the population of rapid and progressive spermatozoa, defined as those moving with velocity VAP > 45 µm/s and coefficient STR > 80%, we observe that, as soon as 3 h of treatment, both bisphenols, BPS and BPA (100 µM), significantly decreased rapid and progressive spermatozoa (Figure 3A) and rapid spermatozoa (Figure 3B) with respect to control but without significant differences between both bisphenols. A lower concentration, 1 µM BPA and BPS, does not cause any effect on these parameters.

In addition, other sperm motility parameters and coefficients evaluated by CASA after bisphenol treatments are shown in Table 1. The short treatment (3 h) of pig spermatozoa with BPA 100 µM causes a significant reduction in all motility parameters except ALH. Treatment with the same concentration of BPS also causes a significant reduction in most of the parameters, except ALH, STR and WOB, whereas no changes were detected after 20 h for any bisphenols in comparison to control.

### 2.2. Effects of BPS and BPA on the Viability of Pig Spermatozoa

In order to evaluate whether treatment with each bisphenol could cause side effects in pig spermatozoa that would lead to cell death, we evaluated sperm viability after BPS and BPA treatment for 3 and 20 h. Figure 4A shows that none of the experimental conditions significantly affected the percentage of SYBR14^+^/PI^−^ sperm, except for BPA at 100 µM for 20 h, which significantly reduced the population of live spermatozoa by 34%.

### 2.3. Effects of BPS and BPA on Pig Spermatozoa Reactive Oxygen Species Content

Figure 5 shows the results of the measurement of sperm ROS content after exposure to bisphenols A and S, showing the relative fluorescence intensity (RFI) of CellROX. Sperm treatment with BPA 100 µM for 3 h (Figure 5A) results in an increase in the cytosol ROS content, which becomes significant at 20 h of incubation (Figure 5B). However, no significant differences are observed after short (3 h) or long-term (20 h) exposure to BPA 1 µM or any BPS concentration (Figure 5).

We further decided to evaluate oxidant species content in the mitochondria, mainly due to reactive nitrogen and oxygen species that can be determined by using a dihydrorhodamine 123 probe. As shown in Figure 6, both bisphenols at any concentration tested cause an increasing time-dependent effect on mitochondrial reactive species. The increase in mitochondrial oxidant molecules is significant after 20 h exposure to any concentration of bisphenol used (Figure 6B).

### 2.4. Effects of BPS and BPA on Spermatozoa Mitochondrial Membrane Potential (MMP)

The population of pig spermatozoa exhibiting high mitochondrial membrane potential (ΔΨm) was evaluated after each bisphenol treatment. As shown in Figure 7A, BPS or BPA treatment does not statistically modify the sperm mitochondrial membrane potential at 3 h. However, 20 h exposure of spermatozoa to BPA 100 µM significantly decreased the sperm population exhibiting high ΔΨm by 3-fold (Figure 7B).

### 2.5. Effects of BPS and BPA on Pig Sperm Plasma Membrane Lipid Organization

In order to study whether bisphenols might cause an impact on the sperm membrane, we analyzed the lipid organization of the plasma membrane. As shown in Figure 8, BPS or BPA treatment does not significantly modify the lipid organization of sperm plasma membrane at 3 h (Figure 8A) or at 20 h (Figure 8B); although lipid disorganization after 20 h treatment with BPA 100 µM tended to be higher, the difference was not significant.

### 2.6. Effects of BPS and BPA on the Intracellular Signaling Pathways Mediated by PKA GSK3α/β and Tyrosine Phosphorylation in Pig Spermatozoa

In order to further study the in vitro effects of BPS and BPA in porcine spermatozoa, we investigated their possible intracellular impact on PKA, GSK3 α/β phosphorylation and tyrosine phosphorylation pathways which are essential for sperm motility and capacitation control.

BPS exposure does not modify the phosphorylation of sperm PKA substrates at any concentration or time studied (Figure 9). However, the treatment of pig spermatozoa with BPA 100 µM significantly reduces the PKA-induced phosphorylation of its downstream substrates, at least those named I to IV in Figure 9, at both 3 h (left panels) and 20 h (right panels). The lower BPA concentration (1 µM) does not cause a significant effect on the phosphorylation of any PKA substrate.

In addition, BPS exposure does not modify GSK3α/β phosphorylation (Figure 10) or tyrosine phosphorylation (Figure 11) in pig spermatozoa at any time or concentration tested. Similarly, no significant differences are observed in GSK3α/β phosphorylation or tyrosine phosphorylation in spermatozoa treated with BPA 1 µM.

However, the exposure to the higher BPA concentration, 100 µM, causes a slight but reproducible reduction in GSK3α phosphorylation that is visible at 3 h (left panels) and 20 h (right panels) without any effect on tyrosine phosphorylation (Figure 11). BPA 100 µM significantly halved GSK3β phosphorylation after 20 h of treatment (Figure 10, right panels).

## 3. Discussion

This work shows that in vitro treatment with BPS or BPA causes a serious adverse effect on the motility of porcine spermatozoa. This negative effect of BPA confirms previous results described in vitro for BPA in the spermatozoa of mammals such as mice [25,26,27,37], bulls [28,38] and humans [30] and in vivo in rats [39,40]. However, discrepancies in BPA effects according to the time and dose/concentration used have been reported in bull spermatozoa [38] where low concentrations of BPA (1–10 µg mL^−1^) clearly increase motility at 24 h. In addition, discrepancies have been reported according to the animal strain, as described in rats [24,39]. Thus, (i) a negative effect on sperm motility caused by in vivo BPA administration [39,40], (ii) no BPA effect on Sprague Dawley rats using 50 mg/kg/day [23] or up to 500 mg/kg/day [41], and (iii) an increase in sperm motility in Wistar rats using 200 mg/kg/day of BPA [24] have been reported. As for BPS, our results showing a clear negative effect on pig sperm motility are in line with those reported in vitro in spermatozoa from humans [22] and bulls [28] and also in vivo in rats (50 µg/L in drinking water) for 10 weeks [40]. However, our finding contrasts with that described by Ullah A et al. (2019) [23] in rats, where BPS did not affect motility probably due to the different way (in vivo), dose (50 mg/kg/day) and time (18 days) of BPS administration, and also contrasts with that described in human spermatozoa [31] where no effect on motility was observed even with higher in vitro BPS concentrations (up to 400 µM).

Apart from the confirmation that the bisphenol molecule exerts a negative action on mammalian sperm motility, this work demonstrates that important differences exist between the bisphenol types, i.e., the radicals present in them, BPA or BPS (sulfone). Thus, bisphenols differ in the intensity of the adverse action on pig sperm motility, which is more visible at shorter times (3 h), where BPA (100 µM) reduces the population of motile spermatozoa by 48% while BPS (100 µM) reduces it by 18%; they also differ in the cellular mechanisms and intracellular pathways triggered.

On the one hand, sperm treatment with BPA (20 h) causes a drastic almost total reduction in sperm motility at 100 µM, a concentration that also reduces sperm viability, although to a lesser extent. The fact that BPA decreases sperm viability in vitro corroborates previous studies in humans [30] and mice [27,37]. Contrary to BPA, the lower reduction in motility caused by BPS, focused on the population of progressive motile and rapid progressive spermatozoa, is not accompanied by any significant BPS effect on sperm viability. This finding in pig sperm viability agrees with data from human spermatozoa [31] where the lack of a BPS effect on viability is clear even at a higher in vitro BPS concentration (400 µM). However, it contrasts with that found in rats [40] where BPS caused a decrease in sperm viability. These discrepancies could be due to the different species studied and to BPS being administered in vivo in rats. The fact that BPS has no effect on pig sperm viability indicates that the lower motility observed in pig spermatozoa is not due to BPS-induced sperm death. On the contrary, BPA at the same time (20 h) and concentration (100 µM) potently reduces cell viability as described in mice [27], which indicates that the radical sulfone added to the bisphenol molecule (BPS) is decisive in the final impact on sperm viability and that intracellular processes are specific to each bisphenol molecule, at least in pig spermatozoa.

Analyzing possible causes of the reduced sperm motility, we studied mitochondrial activity after exposure to BPA and BPS. Again, we found a differential effect between both bisphenols, since at 20 h, only BPA (100 µM) reduces mitochondrial membrane potential, as has been previously reported in rats [26,27]. The BPA-induced mitochondrial dysfunction can help explain the decrease in pig sperm motility caused by this bisphenol. However, as described recently in human spermatozoa [31], BPS (100 µM) does not affect mitochondrial membrane potential in pig spermatozoa, demonstrating that intracellular mechanisms are specific to each bisphenol and also that the BPS-induced reduction in motility is not due to lower mitochondrial activity in pig spermatozoa.

Interestingly, both bisphenols cause an increase in reactive oxidant species in the sperm mitochondria (evaluated by DHR fluorescence) at a concentration and time that cause a reduction in pig sperm motility. Thus, it could be inferred that the decrease in pig sperm motility could be mediated, at least in part, by an increase in reactive oxidizing species in the mitochondria caused by the bisphenol molecule, regardless of the substitution in its side chain (BPA or BPS). This result contrasts with that found in human spermatozoa where BPS at the same concentration did not affect mitochondrial ROS [31], which could be explained by the different species studied. However, when investigating cellular ROS levels, only BPA significantly increases them, as BPS treatment has no effect. In the literature, BPA has been found to increase ROS and malondialdehyde (MDA) levels in mitochondria isolated from mouse spermatozoa, therefore inducing oxidative stress [37], and BPS led to a higher ROS production in rat spermatozoa [23]. On the contrary, it has also been reported that BPA does not cause any effect on sperm ROS content, whereas BPS decreases it [28], although these differences can be explained by the fact that this study was performed in cryopreserved bovine spermatozoa [28]. The idea that the reduction in motility can be due to bisphenol-induced high ROS levels has been previously suggested for BPA and/or BPS in spermatozoa from other species such as humans with BPA [30], mice with BPA [37] and with BPS [25], and rats with BPS [23]. Our results indicate that intracellular mechanisms leading to cellular, but not mitochondrial, ROS production in pig spermatozoa seem to depend on the radical substituted in the bisphenol molecule, as sulfone substitution seems to not lead to an increase in cellular ROS.

Focusing on the plasma membrane of porcine spermatozoa, it seems that the bisphenols do not affect its fluidity, although a tendency to a greater disorganization of the membrane lipids is detected after treatment only with BPA. There are no previous works investigating the effect of bisphenol on sperm membranes, but this finding suggests that the decrease in pig sperm motility triggered by BPS or BPA is not likely due to a negative impact on the plasma membrane.

When looking at intracellular pathways that control sperm motility to investigate bisphenol action, we find a differential effect between BPA and BPS. On one hand, BPS does not affect any of the phosphorylation pathways studied (GSK3β and PKA) in pig spermatozoa at experimental conditions where it clearly reduces motility. To date, there are no previous studies in the literature that investigated intracellular pathways induced by BPS in spermatozoa. Therefore, our results suggest that intracellular mechanisms leading to a reduction in motility triggered by BPS do not include GSK3β or PKA phosphorylation pathways. On the contrary, BPA treatment leads to a significant inhibition of both phosphorylation pathways: GSK3β and at least four protein bands corresponding to downstream PKA substrates. Interestingly, BPA’s interference in Ser/Thr phosphorylation pathways that control pig sperm motility occurs only at conditions where BPA decreases motility, suggesting that the impairment of these signaling pathways can be, at least in part, responsible for the BPA-induced inhibition of pig sperm motility. Our results contrast with those found in mice where BPA (100 µM) activated several transduction pathways including p38 MAP, p85 (PI3K) and PKA [26,27]. In these studies, in mice, the authors also reported BPA effects different than ours in pig spermatozoa, such as a reduction in the mitochondrial membrane potential, suggesting that BPA sperm effects and the intracellular signaling involved seem to depend on the species studied. It has been proposed that bisphenols might directly act on progesterone- or prostaglandin-dependent pathways leading to CatSper activation in human spermatozoa [42,43]. Therefore, we cannot discard that in our experimental conditions in pig spermatozoa, BPA but not its structural analog BPS could be interfering in intracellular pathways depending on progesterone or other ligands involving Ser/Thr phosphorylation, such as PKA and GSK3β.

None of the bisphenols studied modifies pig sperm intracellular signaling pathways based on protein tyrosine phosphorylation, which agrees with the fact that the involvement of protein tyrosine phosphorylation in pig spermatozoa has been clearly demonstrated during the capacitation process, but it is not mediating motility. However, our results contrast with those reported only for BPA, not for BPS, in mice spermatozoa where BPA leads to an increase in tyrosine phosphorylation levels [26,27], which suggests again that BPA sperm effects and intracellular pathways involved seem to depend on the species studied.

In summary, this work shows that bisphenols negatively affect an essential sperm function, motility. However, the intensity of this adverse effect and the intracellular mechanisms involved differ according to the type of bisphenol, that is, the radical present in its lateral chain (BPA or BPS). On the one hand, the reduction in motility caused by BPA, accompanied by lower sperm viability, could be explained by a decrease in mitochondrial membrane potential and an increase in the amount of mitochondrial and cellular reactive oxygen species, cellular events that can affect motility. In addition, this work clearly shows that BPA causes a decrease in the phosphorylation activity of GSK3β and PKA pathways. Given the implication of both signaling pathways in the regulation of porcine sperm motility, these data can also contribute to explaining the strong decline in motility caused by BPA. On the contrary, the lower reduction in motility triggered by BPS does not modify any of the sperm cellular parameters or intracellular pathways affected by BPA, except the increase in mitochondrial reactive molecule species that could, at least partially, contribute to explaining the lower motility. This work suggests that the bulkier side chain in the BPS molecule seems to alter the sperm effects of the bisphenol A molecule. This fact indicates that further research would be important in understanding the sperm intracellular pathways and molecular and cellular mechanisms that are responsible for the decrease in pig sperm motility induced by the BPA analog BPS. The conclusions of this work regarding the impact of BPS on male reproductive health should be considered in the context of the increased daily exposure of animals and humans to products that contain BPS.

## 4. Materials and Methods

### 4.1. Chemical and Sources

Bisphenol A (BPA), 4,4′-Sulfonyldiphenol (BPS), dihydrorhodamine 123 (DHR) and merocyanine M540 were from Sigma-Aldrich (St Louis, MO, USA); propidium iodide (PI), SYBR-14, 5,5′,6,6′–tetrachloro-1,1′,3,3′ tetraethylbenzymidazolyl carbocyanine (JC-1), Yo-Pro-1 and CellROX Green probe were from Thermo Fisher Scientific (Eugene, OR, USA); DC Protein Assays and 2X Laemmli Sample Buffer were from Bio-Rad (Hercules, CA, USA); and Intercept (TBS) blocking buffer and IRDye 800RD and 680RD secondary antibodies were from LI-COR Biotechnology (Bonsai Lab, Alcobendas, Spain). Furthermore, the anti-phospho (Ser/Thr) PKA Substrate (#9624), anti-phospho (Ser 21/9) GSK3α/β (#9331) and anti-total GSK3β (#9332) polyclonal antibodies were from Cell Signaling Technology, Inc. (Beverly, MA, USA); the anti-phospho-tyrosine monoclonal antibody (4G10, 05-321) was from Millipore (Burlington, MA, USA); the anti-α-tubulin monoclonal antibody (TU-02, #SC-8035) was from Santa Cruz Biotechnology (Santa Cruz, CA, USA). All reagents used to prepare incubation media were purchased from Sigma-Aldrich (St. Louis, MO, USA).

### 4.2. Spermatozoa Incubation Media

Tyrode’s basal medium (TBM; 96 mM NaCl, 4.7 mM KCl, 0.4 mM MgSO_4_, 0.3 mM NaH_2_PO_4_, 5.5 mM glucose, 1 mM sodium pyruvate, 21.6 mM sodium lactate, 20 mM HEPES, 5 mM EGTA and 0.02% PVA) was prepared and used as the incubation medium. The medium was prepared on the day of use and adjusted to pH 7.45 with an osmolarity of 290–310 mOsm kg^−1^.

### 4.3. Semen Samples

Sperm samples from Duroc boars (2–4 years old) were commercially obtained from a regional porcine company (Tecnogenext, S.L, Mérida, Spain), without any requirement of approval from the animal research review board of the University of Extremadura. All boars were housed in individual pens in an environmentally controlled building (15–25 °C) according to Regional Government and European regulations and received the same diet. Fresh ejaculates were collected with the gloved hand technique, diluted in a commercial extender, and stored at 17 °C before use in the laboratory.

### 4.4. Boar Sperm Preparation

To minimize individual boar variations, samples from up to 3 animals were pooled using semen from no less than 12 boars in different combinations. Only semen pools with at least 80% morphologically normal spermatozoa were used. Semen was centrifuged at 900× *g* for 4 min and washed with phosphate-buffered saline (PBS), and spermatozoa were placed in TBM medium at a final concentration of 30 × 10^6^ spermatozoa mL^−1^.

Previously, we evaluated the effect of both bisphenols on motility parameters in pig semen by constructing a dose–response curve with different BPA concentrations, 10^−12^ to 10^−4^ M; based on the results, we continued to work with concentrations of 1 μM and 100 μM. For the experimental procedure, 0.5 mL spermatozoa samples containing 30 × 10^6^ spermatozoa mL^−1^ were incubated in TBM for 3 h and 20 h at 38.5 °C with 5% CO_2_ in the absence (control, DMSO 0.3%) or presence of 4,4′-(Propane-2,2-diyl) diphenol (BPA, 1 and 100 μM) or 4,4′Sulfonyldiphenol (BPS, 1 and 100 μM) previously diluted in a final concentration of 0.3% DMSO.

To minimize possible experimental variations, all the different experimental treatments were carried out in each of the semen pools.

### 4.5. Evaluation of Spermatozoa Motility

After incubation with or without BPA or BPS, 2.5 µL spermatozoa samples were placed in a 38.5 °C pre-warmed counting chamber with 20 μm depth (Leja, Nieuw-Vennep, The Netherlands). Spermatozoa images were taken using a microscope equipped with a 10× negative-phase contrast objective, with a heated stage, and a CCD camera that takes 25 consecutive digitalized images obtained during 1 s from at least 3 different fields and 300 spermatozoa per sample. Digitalized images were analyzed using a computer-assisted semen analysis system, specifically the ISAS system (Integrated Semen Analysis System, Proiser R + D, Paterna, Valencia, Spain). Sperm motility parameters and coefficients were as follows: motile spermatozoa (percentage of spermatozoa with an average path velocity > 10 μm/s), rapid spermatozoa (percentage of spermatozoa with an average path velocity > 45 µm/s), progressive motile spermatozoa (percentage of spermatozoa with a straightness coefficient > 80%), rapid and progressive spermatozoa (VAP > 45 µm/s and STR > 80%), VCL (curvilinear velocity in μm/s), VSL (straight-line velocity in μm/s), VAP (average path velocity in μm/s), LIN (linearity coefficient in %), STR (straightness coefficient in %), WOB (wobble coefficient in %), ALH (amplitude of lateral head movement in µm) and BCF (beat cross of flagellum frequency).

### 4.6. Flow Cytometry Analysis

Flow cytometry analysis was performed using an BD Accuri C6 Plus flow cytometer (Becton Dickinson and Company, BD Biosciences, San José, CA, USA) equipped with three detection channels for a blue laser (488 nm), namely FL-1 (533/30 nm bandpass filter), FL-2 (585/40 nm bandpass filter) and FL-3 (670 LP.H. nm bandpass filter), and a detection channel for a red laser (640 nm): FL-4 (675/25 nm bandpass filter). Flow cytometry experiments and data analysis, including relative fluorescence intensity (RFI) expressed in arbitrary units, were performed using BD Accuri C6 software (Becton Dickinson and Company, BD Biosciences, CA, USA).

#### 4.6.1. Analysis of Spermatozoa Viability by Flow Cytometry

Fluorescent staining using SYBR-14 and propidium iodide (PI) was performed to measure sperm viability as described previously [44]. Briefly, 5 µL of SYBR-14 (2 μM) and 10 µL of PI (240 μM) were added to 50 µL of spermatozoa (30 × 10^6^ cells mL^−1^) diluted with 450 µL of TBM until final concentrations of 20 nM for SYBR-14 and 5 μM for PI were reached. Then, the samples were incubated for 15 min at room temperature (RT) in darkness and analyzed in the flow cytometer. Results of viable spermatozoa were expressed as the percentage of SYBR-14+ and PI− spermatozoa.

#### 4.6.2. Analysis of Sperm Mitochondrial Membrane Potential (ΔΨm) by Flow Cytometry

Mitochondrial membrane potential variations (ΔΨm) were evaluated using the specific probe JC-1 (5,5′,6,6′–tetrachloro-1,1′,3,3′ tetraethylbenzymidazolyl carbocyanine iodine) as described previously [44].

The experimental procedure consists of diluting 50 μL of spermatozoa (30 × 10^6^ cells mL^−1^) in 450 µL of TBM containing 15 μM of JC-1, mixed and incubated at 38.5 °C for 30 min. The fluorescence values were recorded on channels FL-1 (JC-1 monomer) and FL-2 (JC-1 polymer or high mitochondrial membrane potential). Results were expressed as the percentage of spermatozoa with high mitochondrial membrane potential (high ΔΨm) with respect to the total number of spermatozoa analyzed.

#### 4.6.3. Analysis of the Degree of Sperm Plasma Membrane Lipid Organization by Flow Cytometry

Fluorescent staining using the probe merocyanine M540 as a membrane lipid fluidity marker and the probe Yo-Pro-1 as a marker of changes in plasma membrane permeability (commonly associated with cell death) was performed as described previously [44] Briefly, 100 µL of spermatozoa (30 × 10^6^ cells mL^−1^) was diluted in 400 µL of TBM containing 75 nM of Yo-Pro-1 and 6 μM of M540 and incubated at 38.5 °C for 10 min. Labeled spermatozoa were categorized as (i) viable cells with low plasma membrane lipid disorder (M540/YoPro-1^−^), (ii) viable cells with high plasma membrane lipid disorder (M540^+^/YoPro-1^−^) or (iii) non-viable cells (Yo-Pro-1^+^). Results were expressed as the mean relative fluorescence intensity (RFI) of M540^+^ in live spermatozoa (Yo-Pro-1^−^).

#### 4.6.4. Evaluation of Reactive Oxygen Species Production in Pig Spermatozoa

The reactive oxygen species production was evaluated using specific probes: CellROX, for reactive oxygen species, and DHR, for reactive oxygen and nitrogen species. Briefly, 30 µL of spermatozoa (30 × 10^6^ cells mL^−1^) diluted with 470 µL of TBM, until a final concentration of 5 μM was reached for CellROX, was incubated for 30 min at 38.5 °C. The fluorescence values were calculated based on the RFI of CellROX. Moreover, 30 µL of spermatozoa (30 × 10^6^ cells mL^−1^) was diluted with 470 µL of TBM, until a final concentration of 1 μM was reached for DHR. Then, the samples were incubated for 20 min at 38.5 °C in darkness and analyzed in the flow cytometer. The fluorescence values were calculated based on the RFI of DHR. For both, results were expressed as the mean RFI in arbitrary units.

### 4.7. Analysis of Pig Spermatozoa Phosphorylated Proteins by Western Blotting

Spermatozoa (0.5 mL) were centrifuged at 5000× *g* for 1 min at RT, washed in PBS medium and centrifuged again. The pellet was resuspended in 30 µL of Laemmli Sample Buffer (2×), incubated for 10 min in constant rotation and then centrifuged at 10,000× *g* for 10 min at 4 °C. The protein concentration of the supernatant was determined using a Bio-Rad DC Protein Assay. After protein concentration analysis, 2-mercaptoethanol (2.5% *v*/*v*) was added to the sperm lysates before heating for 5 min at 95 °C and storage at −20 °C.

Sperm proteins (10 μg) were resolved using 10% SDS-PAGE. Electrophoresis was run at 90 V for the first 20 min and then 145 V for another 90 min at room temperature in 1× running buffer. After electrophoresis, proteins were transferred to nitrocellulose membranes at 380 mA for 2.5 h and then were blocked for 1 h using Intercept (TBS) blocking buffer containing 0.2% Tween-20. Membranes were then incubated at 4 °C overnight using anti-phospho-GSK3α/β (1:1000), anti-GSK3β (1:1000), anti-phospho-PKA-substrates (1:1000), anti-phospho-tyrosine (1:1000) or anti-α-tubulin (1:5000) antibodies.

The membranes were then washed and incubated with the appropriate secondary antibody IRDye 800RD or 680RD as indicated by the manufacturer. Fluorescence was detected using an Odyssey Fc Imaging System (LI-COR Biotechnology), and all bands were quantified using the Image Studio software from LI-COR.

### 4.8. Statistical Analysis

To determine if the differences between treatments were statistically significant, hypothesis tests were carried out. Data were analyzed for normal distribution with a Kolmogorov–Smirnov test and for homoscedasticity with a Levene test. Differences were determined by a parametric test, namely a one-way analysis of variance (ANOVA) followed by post hoc Tukey. In Figure 1, Figure 2, Figure 3, Figure 4, Figure 5, Figure 6, Figure 7 and Figure 8, results are expressed in box-and-whisker plots which show the distribution of data into quartiles, the lines extending from the box (whiskers) indicating variability outside the upper and lower quartiles. Inside the box, the transverse line represents the median percentile, and the “x” symbol denotes the mean. In Figure 9, Figure 10 and Figure 11, data are shown as the mean ± standard error of the mean (SEM). All analyses were performed using SPSS v27 for Windows software (SPSS Inc. Chicago, IL, USA). Statistical significance was set at a *p* value lower than 0.05.

## Figures and Tables

**Figure 1 ijms-24-09598-f001:**
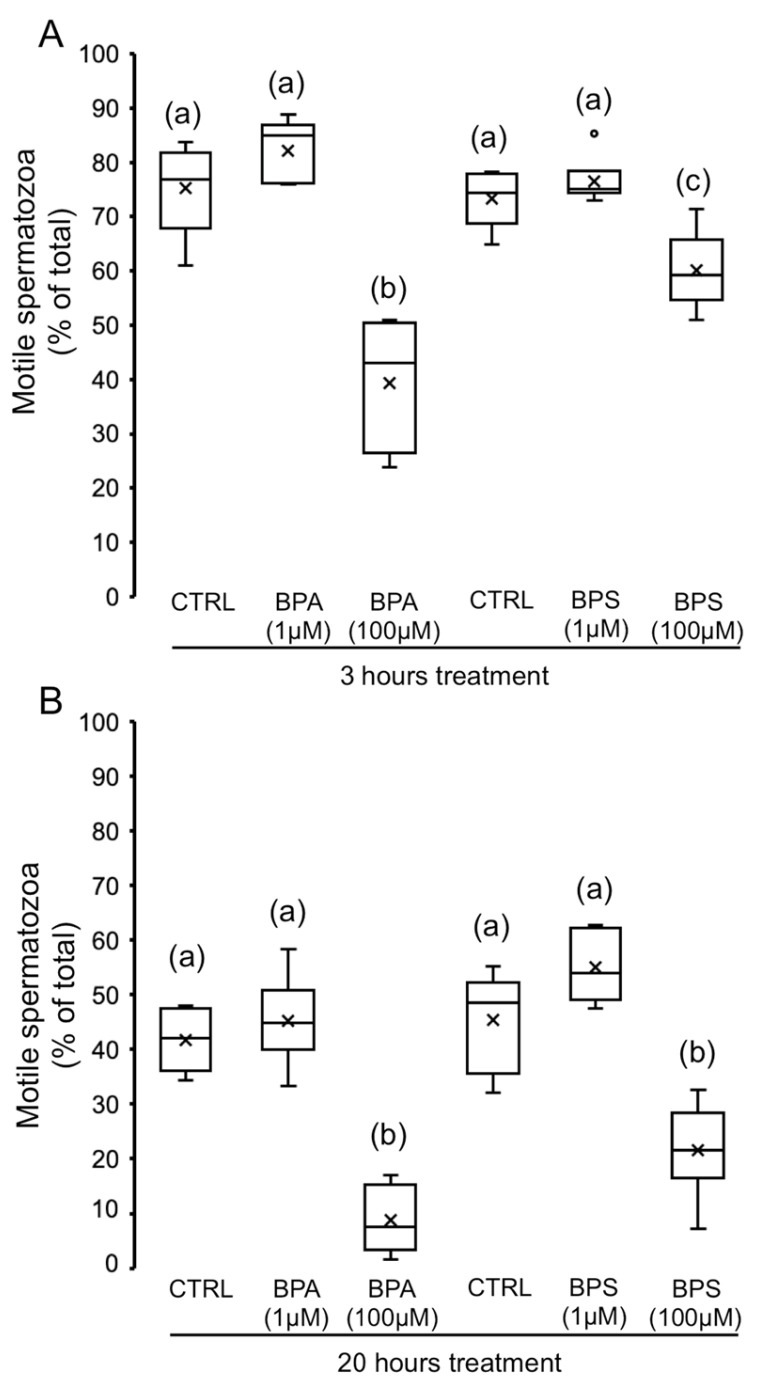
Effects of BPS and BPA on pig spermatozoa motility. Pig spermatozoa were incubated in TBM at 38.5 °C with 5% CO_2_ in the absence (control) or presence of BPA (1 and 100 μM) or BPS (1 and 100 μM). The percentage of motile spermatozoa was evaluated at 3 h (**A**) (F = 22.171; *p* < 0.001) and 20 h (**B**) (F = 32.106; *p* < 0.001), and the results are depicted in box-and-whisker plots. The whiskers extend to the largest and smallest data points; the box extends from the upper quartile to the lower quartile and is crossed by a line at the median of the data. Circles represent outliers. Each experiment was performed 6 times (*n* = 6). Boxes with different letters are statistically different from each other, *p* < 0.05.

**Figure 2 ijms-24-09598-f002:**
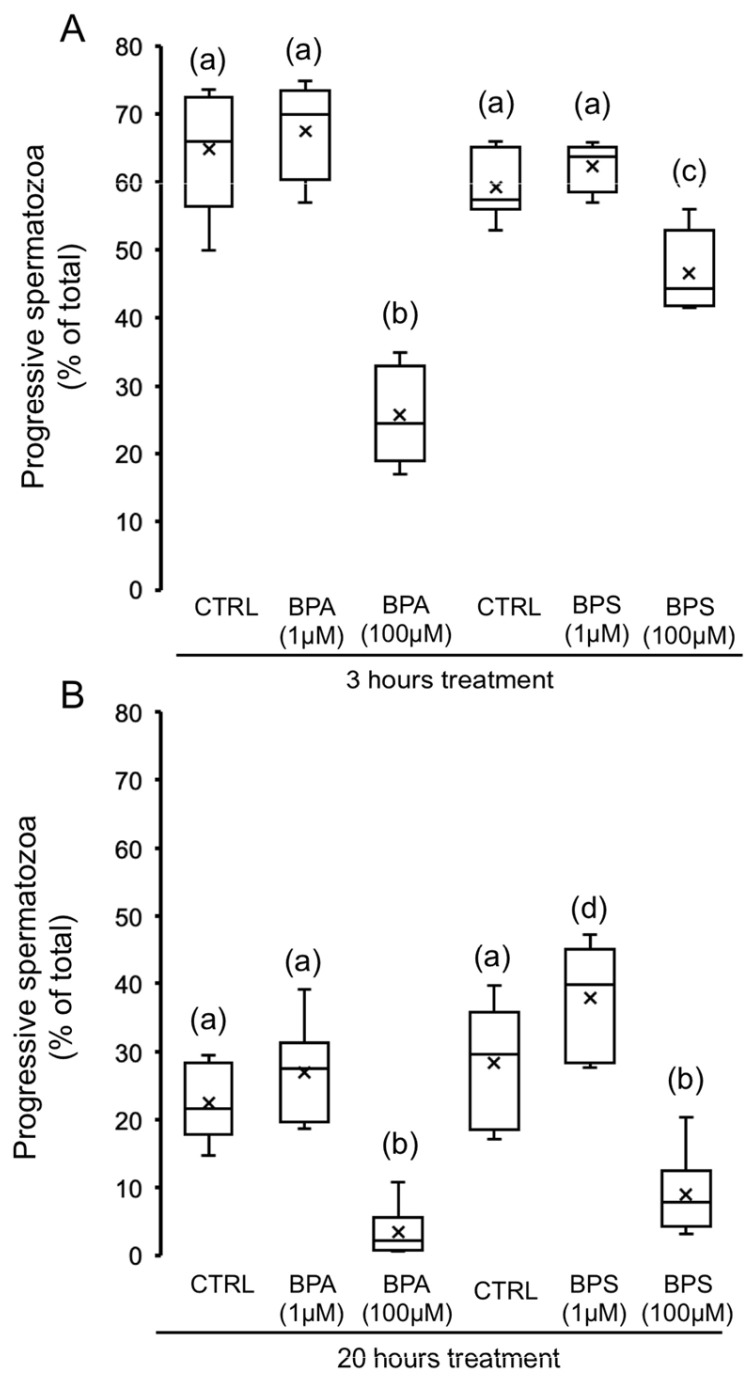
Impact of BPS and BPA on progressive motility. Pig spermatozoa were incubated in TBM at 38.5 °C with 5% CO_2_ in the absence (control) or presence of BPA (1 and 100 μM) or BPS (1 and 100 μM). The percentage of progressive spermatozoa was evaluated at 3 h (**A**) (F = 30.376; *p* < 0.001) and 20 h (**B**) (F = 20.819; *p* < 0.001) (*n* = 6) and depicted in box-and-whisker plots. The whiskers extend to the largest and smallest data points; the box extends from the upper quartile to the lower quartile and is crossed by a line at the median of the data. Boxes with different letters are statistically different from each other, *p* < 0.05.

**Figure 3 ijms-24-09598-f003:**
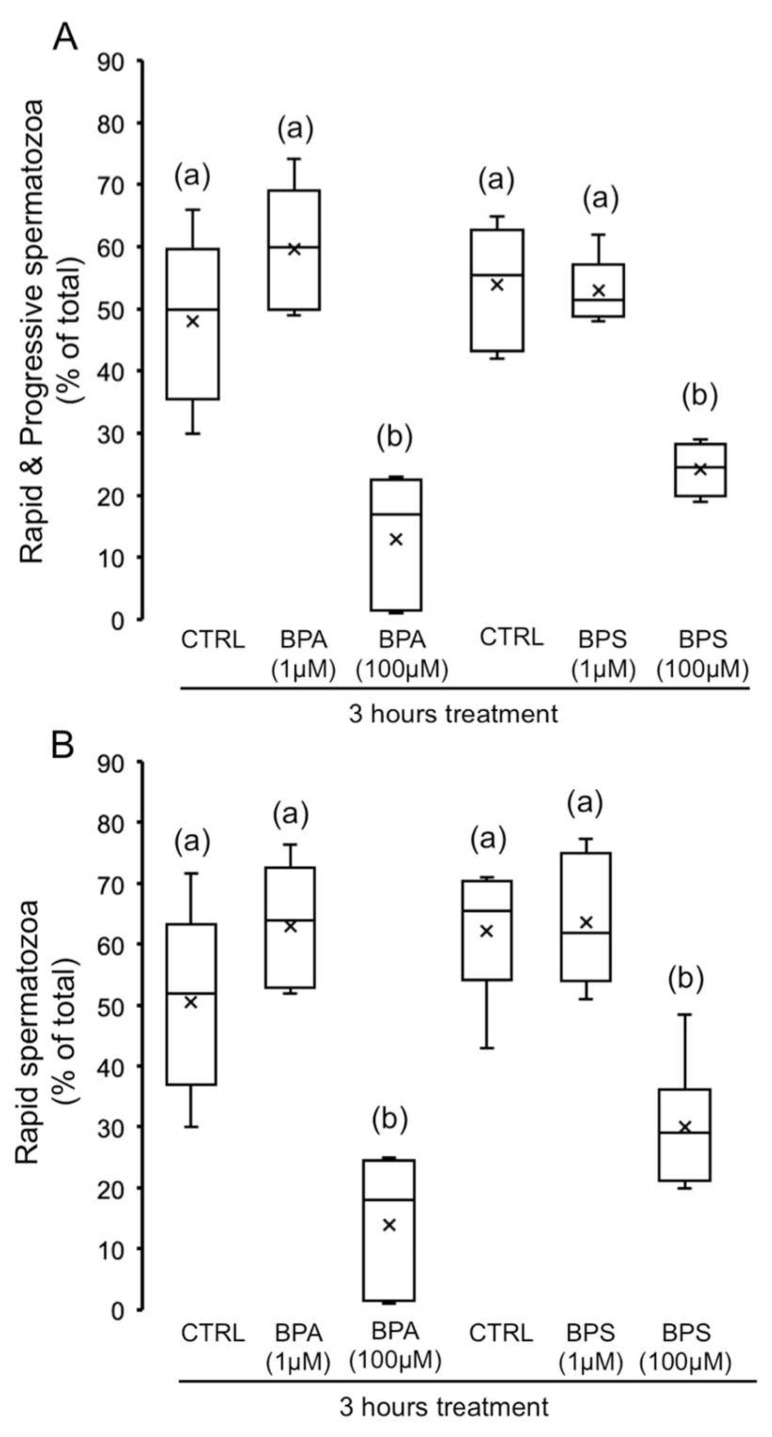
Impact of BPS and BPA on rapid and progressive spermatozoa. Pig spermatozoa were incubated in TBM at 38.5 °C for 3 h with 5% CO_2_ in the absence (control) or presence of BPA (1 and 100 μM) or BPS (1 and 100 μM). Each experiment was performed 6 times (*n* = 6), and the percentages of rapid and progressive (**A**) (F = 22.436; *p* < 0.001) and rapid spermatozoa (**B**) (F = 17.573; *p* < 0.001) are depicted in box-and-whisker plots. The whiskers extend to the largest and smallest data points; the box extends from the upper quartile to the lower quartile and is crossed by a line at the median of the data. Boxes with different letters are statistically different from each other, *p* < 0.05.

**Figure 4 ijms-24-09598-f004:**
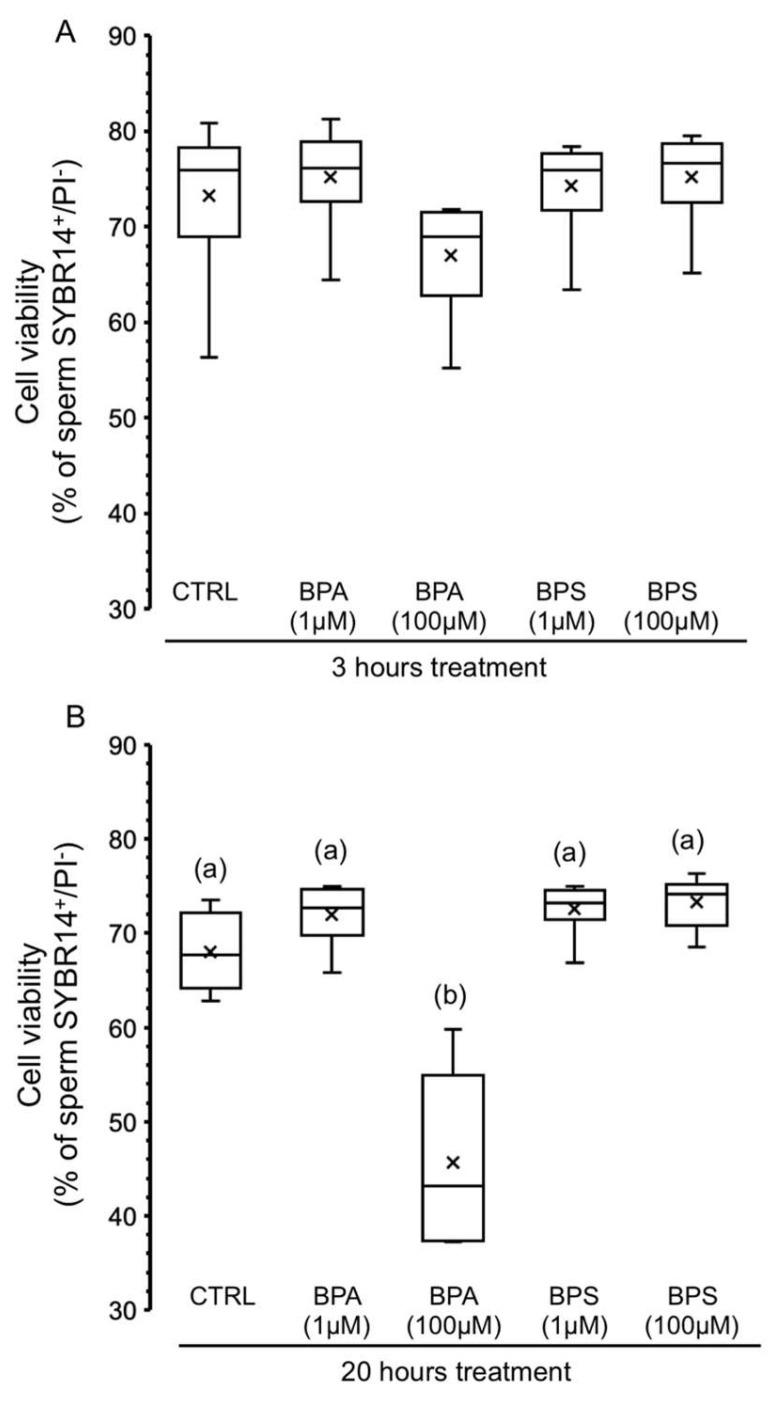
Effects of BPS and BPA on the viability of pig spermatozoa. Pig spermatozoa were incubated in TBM at 38.5 °C with 5% CO_2_ in the absence (control) or presence of BPA (1 and 100 μM) or BPS (1 and 100 μM). This experiment was performed 6 times (*n* = 6), and the percentages of SYBR14-positive and PI-negative spermatozoa are depicted in box-and-whisker plots for 3 h treatment (**A**) (F = 1.753; *p* = 0.170) and 20 h (**B**) (F = 32.372; *p* < 0.001). The whiskers extend to the largest and smallest data points; the box extends from the upper quartile to the lower quartile and is crossed by a line at the median of the data. Boxes with different letters are statistically different from each other, *p* < 0.05.

**Figure 5 ijms-24-09598-f005:**
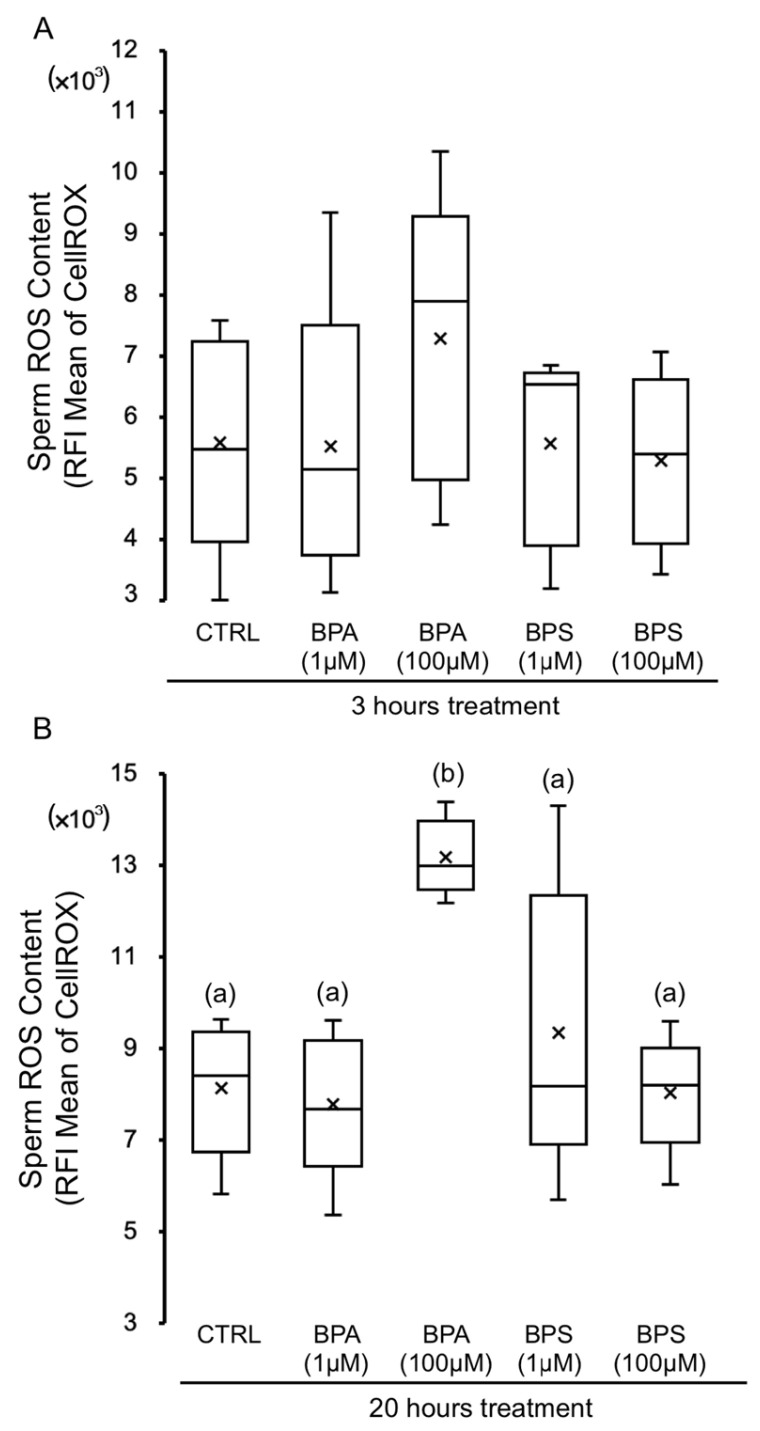
Effects of BPS and BPA on reactive oxygen species (ROS) content of pig spermatozoa. Pig spermatozoa were incubated in TBM at 38.5 °C with 5% CO_2_ in the absence (control) or presence of BPA (1 and 100 μM) or BPS (1 and 100 μM). This experiment was performed 5 times (*n* = 5), and the means of relative fluorescence intensity (RFI) of CellROX-positive spermatozoa are expressed in box-and-whisker plots for 3 h treatment (**A**) (F = 0.873; *p* = 0.150) and 20 h (**B**) (F = 7.224; *p* < 0.001). The whiskers extend to the largest and smallest data points; the box extends from the upper quartile to the lower quartile and is crossed by a line at the median of the data. Boxes with different letters are statistically different from each other, *p* < 0.05.

**Figure 6 ijms-24-09598-f006:**
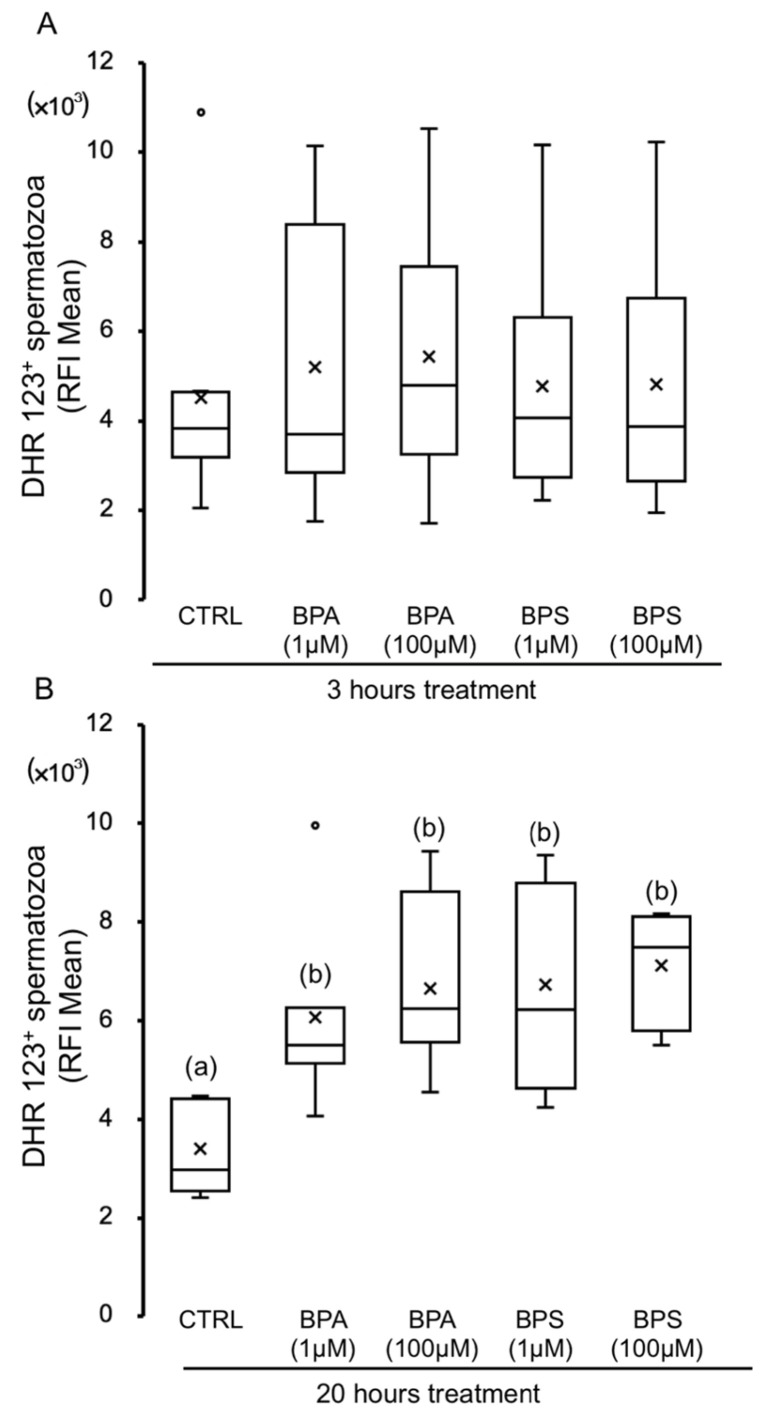
Effects of BPS and BPA on dihydrorhodamine 123 (DHR) oxidation of pig spermatozoa. Pig spermatozoa were incubated in TBM at 38.5 °C with 5% CO_2_ in the absence (control) or presence of BPA (1 and 100 μM) or BPS (1 and 100 μM). Each experiment was performed 7 times (*n* = 7). The mean relative fluorescence intensity (RFI) of DHR was evaluated at 3 h (**A**) (F = 0.156; *p* = 0.950) and 20 h (**B**) (F = 6.209; *p* < 0.001), and the results are depicted in box-and-whisker plots, where whiskers extend to the largest and smallest data points and the box extends from the upper quartile to the lower quartile and is crossed by a line at the median of the data. Circles represent outliers. Boxes with different letters are statistically different from each other, *p* < 0.05.

**Figure 7 ijms-24-09598-f007:**
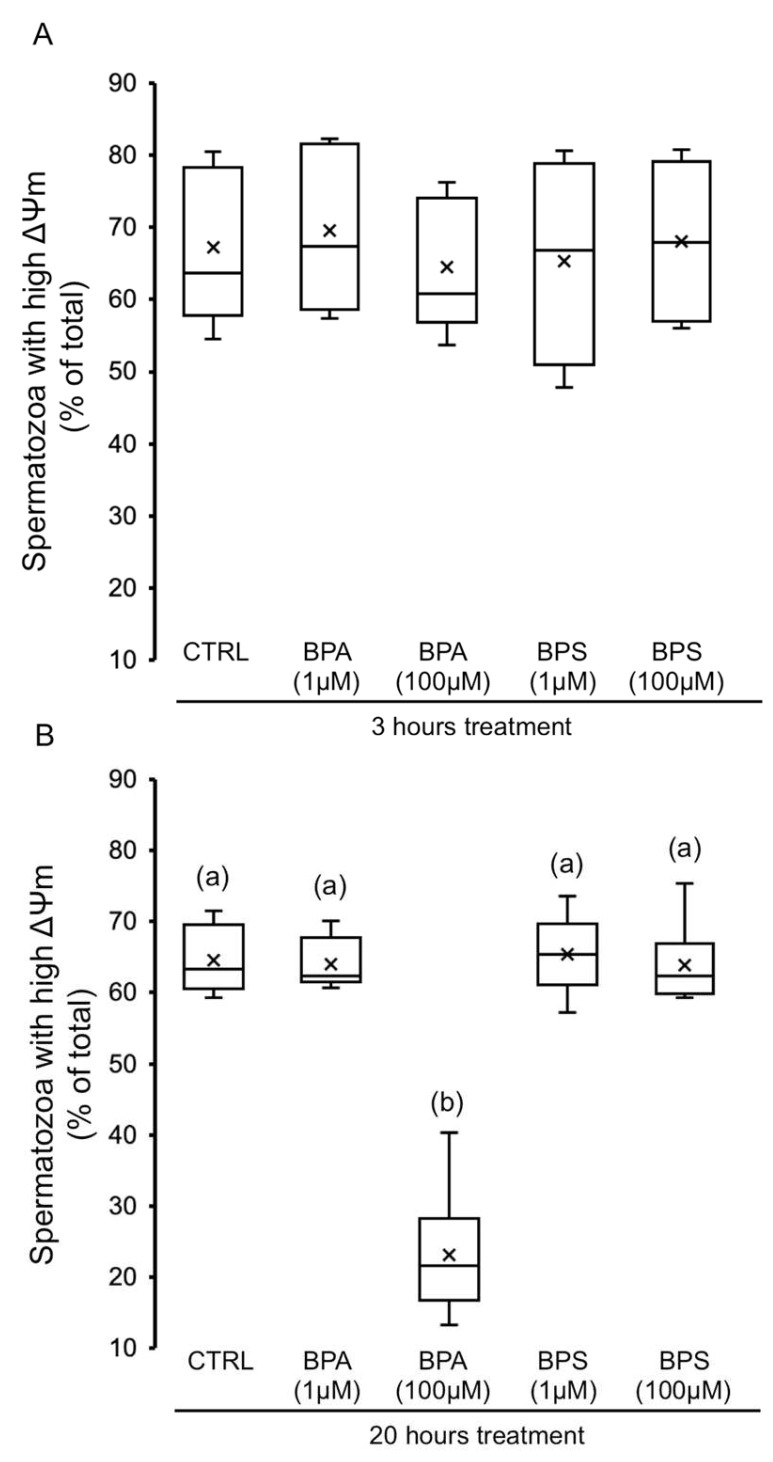
Effects of BPS and BPA on spermatozoa mitochondrial membrane potential (ΔΨm). Pig spermatozoa were incubated in TBM at 38.5 °C with 5% CO_2_ in the absence (control) or presence of BPA (1 and 100 μM) or BPS (1 and 100 μM). Each experiment was performed 6 times (*n* = 6), and the percentage of spermatozoa exhibiting relatively higher ΔΨm was evaluated at 3 h (**A**) (F = 0.158; *p* = 0.950) and 20 h (**B**) (F = 53.844; *p* < 0.001). The results are depicted in box-and-whisker plots. The whiskers extend to the largest and smallest data points; the box extends from the upper quartile to the lower quartile and is crossed by a line at the median of the data. Boxes with different letters are statistically different from each other, *p* < 0.05.

**Figure 8 ijms-24-09598-f008:**
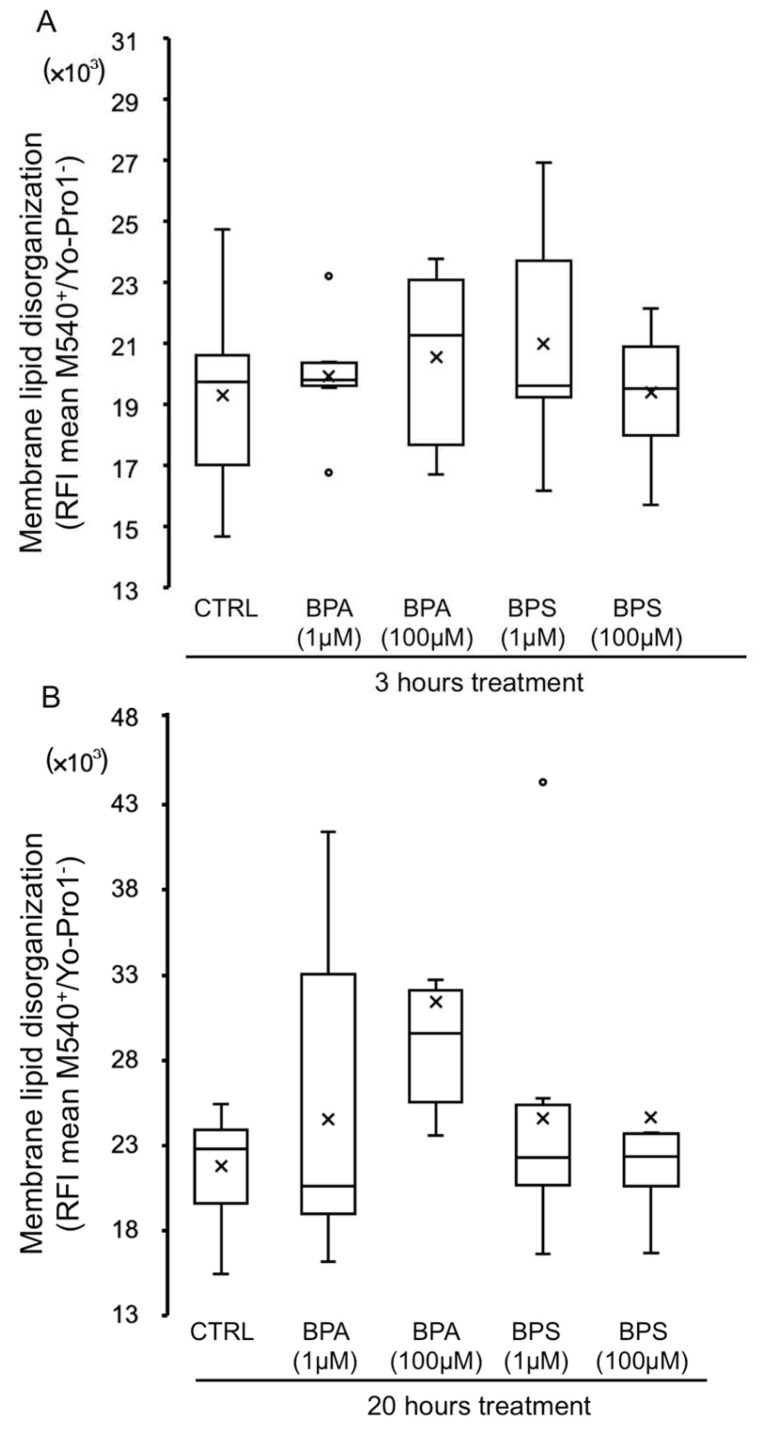
Effects of BPS and BPA on pig sperm plasma membrane lipid organization. Pig spermatozoa were incubated in TBM at 38.5 °C with 5% CO_2_ in the absence (control) or presence of BPA (1 and 100 μM) or BPS (1 and 100 μM). This experiment was performed 8 times (*n* = 8). The mean relative fluorescence intensity (RFI) of M540-positive/Yo-Pro-1-negative spermatozoa is depicted in box-and-whisker plots for 3 h (**A**) (F = 0.598; *p* = 0.660) and 20 h (**B**) (F = 1.506; *p* = 0.220). The whiskers extend to the largest and smallest data points; the box extends from the upper quartile to the lower quartile and is crossed by a line at the median of the data. Circles represent outliers. Boxes with different letters are statistically different from each other, *p* < 0.05.

**Figure 9 ijms-24-09598-f009:**
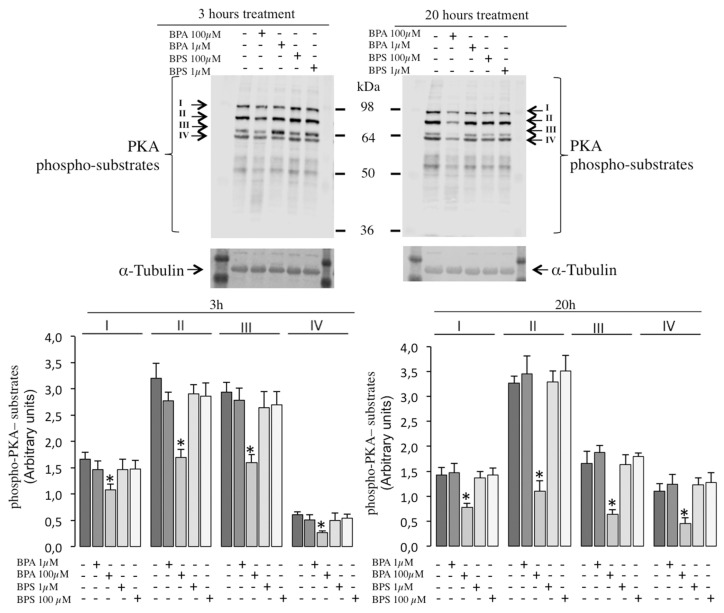
Effects of BPS and BPA on the phosphorylation of intracellular signaling pathways mediated by PKA in pig spermatozoa. Pig spermatozoa were incubated in TBM at 38.5 °C for 3 and 20 h with 5% CO_2_ in the absence (control) or presence of BPA (1 and 100 μM) or BPS (1 and 100 μM). Upper panel: Sperm proteins (10 μg) were analyzed by Western blotting using anti-phospho-PKA substrates as primary antibodies. Each experiment was performed 6 times, and representative films are shown. Loading controls using anti-α-tubulin antibody (lower films) were performed for each experiment in the same membrane. Arrows indicate cross-reactive bands (Bands I–IV) of sperm phosphorylated proteins that are substrates of PKA. Lower panel: Densitometry analysis of Bands I–IV is shown, bars in colors represent the different treatment in each case and values are expressed as mean ± SEM of arbitrary units. Band I (F = 1.831; *p* = 0.150), Band II (F = 7.188; *p* < 0.001), Band III (F = 5.292; *p* < 0.05), Band IV (F = 3.257; *p* < 0.05) for 3 h treatment; Band I (F = 4.533; *p* < 0.05), Band II (F = 15.792; *p* < 0.001), Band III (F = 9.749; *p* < 0.001), Band IV (F = 3.427; *p* < 0.05) for 20 h treatment. Statistical differences are shown with * (*p* < 0.05).

**Figure 10 ijms-24-09598-f010:**
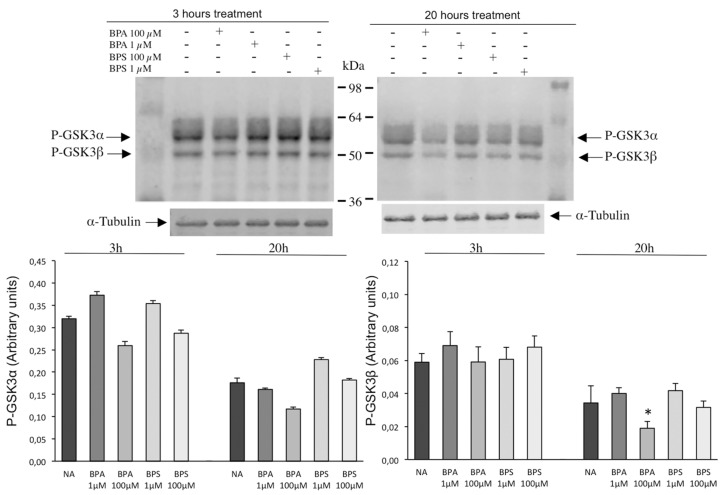
Effects of BPS and BPA on the intracellular signaling pathways mediated by GSK3α/β in pig spermatozoa. Pig spermatozoa were incubated in TBM at 38.5 °C for 3 and 20 h with 5% CO_2_ in the absence (control) or presence of BPA (1 and 100 μM) or BPS (1 and 100 μM). Upper panel: Sperm proteins (10 μg) were analyzed by Western blotting using anti-phospho GSK3α/β as primary antibody. Arrows indicate the reactive sperm bands corresponding to phosphorylated forms of GSK3α (upper arrow) and GSK3β (lower arrow). Each experiment was performed 6 times, and representative films are shown. Loading controls using α-tubulin antibodies (lower films) were performed for each experiment. Lower panel: Densitometry analysis of P-GSK3α and P-GSK3β bands is shown, bars in colors represent the different treatment in each case and values are expressed as the mean ± SEM of arbitrary units. P-GSK3α (F = 1.555; *p* = 0.210) and P-GSK3β (F = 0.435; *p* = 0.780) at 3 h treatment and P-GSK3α (F = 0.513; *p* = 0.720) and P-GSK3β (F = 4.713; *p* < 0.05) at 20 h treatment. Statistical differences are shown with * (*p* < 0.05).

**Figure 11 ijms-24-09598-f011:**
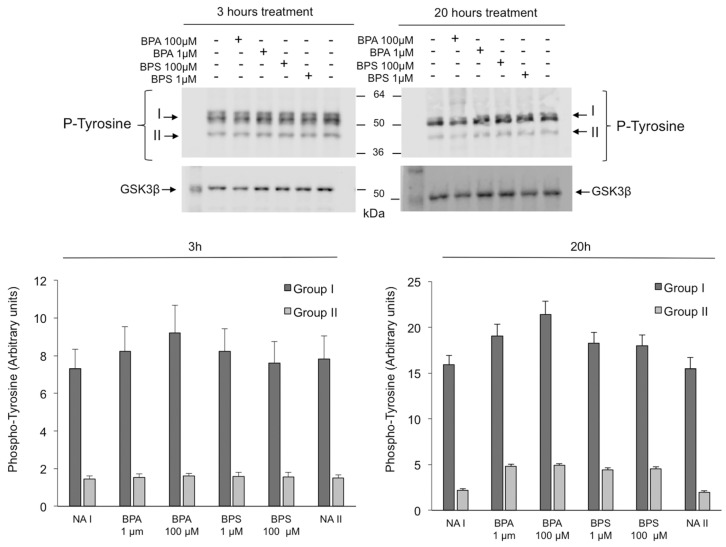
Effects of BPS and BPA on the tyrosine phosphorylation of intracellular signaling pathways in pig spermatozoa. Pig spermatozoa were incubated in TBM at 38.5 °C for 3 and 20 h with 5% CO_2_ in the absence (control) or presence of BPA (1 and 100 μM) or BPS (1 and 100 μM). Upper panel: Sperm proteins (10 μg) were analyzed by Western blotting using anti-phospho-tyrosine as primary antibody. Arrows indicate cross-reactive bands (Bands I and II) of sperm tyrosine-phosphorylated proteins. Each experiment was performed 6 times, and representative films are shown. Loading controls using GSK3β antibodies (lower films) were performed for each experiment. Lower panel: Densitometry analysis of Bands I (referred as Group I) and II (referred as Group II) and II is shown, and values are expressed as the mean ± SEM of arbitrary units. Band I (F = 2.394; *p* = 0.070) and Band II (F = 0.992; *p* = 0.430) at 3 h treatment and Band I (F = 2.951; *p* < 0.05) and Band II (F = 2.314; *p* = 0.080) at 20 h treatment.

**Table 1 ijms-24-09598-t001:** Effects of BPS and BPA on pig sperm motility descriptors.

Time	Treatment	[]	VCL (µm s^−1^)	VSL (µm s^−1^)	VAP (µm s^−1^)	LIN (%)	STR (%)	WOB (%)	ALH (µm)	BCF (Hz)
3 h	BPA	Ctrl	57.9 ± 5.9	44.6 ± 5.3	46.3 ± 5.3	73.8 ± 1.0	91.4 ± 0.6	78.5 ± 0.8	1.9 ± 0.1	8.5 ± 0.3
		1 µM	63.9 ± 3.1	49.1 ± 3.0	50.8 ± 2.7	72.9 ± 1.2	90.4 ± 0.9	77.8 ± 0.7	2.1 ± 0.1	8.8 ± 0.2
		100 µM	41.5 ± 4.3 *	24.2 ± 3.0 *	27.8 ± 3.0 *	57.9 ± 1.4 *	81.8 ± 1.3 *	68.2 ± 1.5 *	1.8 ± 0.1	6.6 ± 0.6 *
	BPS	Ctrl	64.4 ± 3.1	48.8 ± 2.9	51.4 ± 2.9	71.97 ± 0.8	89.2 ± 0.6	77.6 ± 0.6	2.0 ± 0.1	9.0 ± 0.1
		1 µM	63.2 ± 2.3	47.9 ± 2.3	50.3 ± 2.3	72.8 ± 1.2	89.9 ± 0.6	78.1 ± 0.9	2.0 ± 0.1	9.1 ± 0.1
		100 µM	46.9 ± 2.6 *	32.3 ± 2.1 *	35.1 ± 2.1 *	66.6 ± 1.1 *	87.1 ± 0.8	74.4 ± 0.7	1.85 ± 0.1	7.1 ± 0.2 *
20 h	BPA	Ctrl	31.4 ± 1.3	14.4 ± 1.0	18.1 ± 0.8	47.6 ± 1.8	77.3 ± 1.5	60.3 ± 1.7	1.5 ± 0.0	6.1 ± 0.2
		1 µM	35.5 ± 1.0	16.1 ± 0.8	19.6 ± 0.7	46.9 ± 2.2	79.8 ± 1.1	57.6 ± 2.2	1.7 ± 0.0	6.8 ± 0.2
		100 µM	27.3 ± 1.5	11.2 ± 0.9	15.3 ± 0.8	43.0 ± 2.5	70.4 ± 2.6	58.9 ± 2.5	1.5 ± 0.0	4.2 ± 0.6
	BPS	Ctrl	33.22 ± 1.5	16.5 ± 1.6	19.7 ± 1.5	50.3 ± 3.3	80.7 ± 1.5	61.0 ± 3.0	1.5 ± 0.0	6.3 ± 0.3
		1 µM	36.9 ± 2.1	19.0 ± 1.6	22.1 ± 1.5	51.7 ± 2.0	83.1 ± 1.2	60.9 ± 1.8	1.7 ± 0.0	7.4 ± 0.3
		100 µM	28.3 ± 1.0	11.6 ± 0.4	15.6 ± 0.4	43.2 ± 1.9	73.9 ± 1.7	57.7 ± 1.3	1.4 ± 0.0	5.2 ± 0.1

Pig spermatozoa were incubated in TBM at 38.5 °C for 3 and 20 h (*n* = 6) in the absence (Ctrl) or presence of BPA (1 and 100 μM) or BPS (1 and 100 μM). Sperm kinematic parameters were evaluated using an ISAS system: curvilinear velocity (VCL, µm s^−1^), linear velocity (VSL, µm s^−1^), average mean velocity (VAP, µm s^−1^), linearity coefficient (LIN, %), straightness coefficient (STR, %), wobble movement coefficient (WOB, %), mean lateral head displacement (ALH, μm), frequency of head displacement (BCF, Hz). The values are expressed as the mean ± standard error of the mean (SEM). Statistical differences from control are shown with * (*p* < 0.05).

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
