# Peer review of "Bisphenol S Reduces Pig Spermatozoa Motility through Different Intracellular Pathways and Mechanisms than Its Analog Bisphenol A"

_ijms, 2023, doi:10.3390/ijms24119598_

Round 1

Reviewer 1 Report

The article is written clearly. The abstract is informative, and the introduction provides sufficient data to justify the study's purpose. The experimental design is appropriate, and the discussion of results is well-performed.

My only concerns are as follows:

  1. In the introduction, you could provide some numbers to visualize the differences (such as fold changes) between BPA and BPS.
  2. The design of the motility experiment is not perfect, as you have to compare two controls (separately for BPA and BPS). It would be clearer if one control was applied for all analyses (as it is in further experiments). In the case of different controls, you can try to use fold changes to provide values that are comparable between experimental groups (BPA and BPS-treated samples).

Overall, it is a good article and worth being published.

Author Response

Response 1.-As requested, the suggested information to better visualize differences between BPA and BPS has been included in the Introduction section (after former lines 69-72). Specifically, the following sentences are now included: 
“In one hand, BPS did not significantly affect any human sperm parameter, whereas BPA lead to 100% of immotile spermatozoa, 80% decrease in viability and 90% reduction of high mitochondrial membrane potential [31]. On the other hand, in bovine sperm, treatment with both bisphenols reduced progressive motility but with different extent, while BPS reduced it by 45%, BPA decreased it by 60% [28]. However, other study in spermatozoa and oocytes from same species has found that BPA caused a higher oxidative stress than BPS or PBF, suggesting that they are likely acting through different mechanisms [28]”.

Response 2.- We consider this an interesting observation. However, we would like to clarify that both controls shown in the sperm motility Figures (1, 2 and 3) are actually the same control sample (spermatozoa untreated) that was prepared and analyzed in duplicate. In fact, as expected, there are not statistical differences between both duplicates of the control.
In other words, both controls can work/be used for each one of the bisphenols. 
We would like to mention that each one of the treatments shown in Figures 1, 2 and 3 were technically prepared from the same sperm pool and that all of them were analyzed in the same experiment.

Reviewer 2 Report

The study by Mercedes Torres-Badia et al. aims at improving our knowledge on the effect of Bisphenol molecule and related analogues, on reproductive funcion. Authors used pig sperm cells to study in vitro toxicity on spermatozoa. The study is well presented, scirntifically sound and deserves publication.

Author Response

No concerns

Reviewer 3 Report

The authors explore the effect of Bisphenol -A and an alternative suggested healthier replacement, BPS, on in vitro effect on porcine sperm motility parameters, membrane viability, oxidative stress, and potential molecular mechanisms of reduction in motility.   The experimental design is comprehensive and  well designed and demonstrates the negative effects of BPA and BPS on pig sperm motility parameters as well as the difference in the effects of intercellular signaling pathways by the two molecules. The work is well introduced and concluded in connection to other research and how and why further investigation of the impact of Bisphenol molecules on porcine sperm parameters is important for animal and human health  impacts of daily bisphenol exposure.  Although, in the introduction or Methods a reference to why 1 µM and 100 µM and 3 hours and 20 hours were chosen for the treatment concentrations and exposure time would be beneficial- it is touched upon that it was due to a dose-dependent curve, but this could be expanded a little. The chose of sperm parameter variables to measure and why as well as the presentation of the data on the graphs are easy to follow and well presented.

I noted that additional proofreading and grammar check throughout the manuscript would strengthen the work. I noted several spots with an extra word or incorrect negatives. The results could be improved with additional English editing.  Otherwise, the work is well done and well presented.

1)      Grammar- a few examples below

Example Line 55-57

As mentioned above, BPS was introduced in the market as a potentially safer alternative to BPA, although, as well as another BPA analogue, as BPF, is currently no not regulated and the tolerable dose intake has not been identified yet [14].

Line 57 I suggest changing “Moreover” to “However”

Line 55-68 remove the repetitive transition/connectors words – recently—author information may be best put in the cite reference.

Line 70 “or” should be “and”

2)      Results:

Please add Mean and F-test and p-values of the compared variables for the different results sections. If written - an effect is “significant” --the addition of means and  f-test and p-value are needed and helpful to describe results.

I like the addition of the %change to explain the changes but take in account that the variables are mostly percentage variables so this could be confusing and may be good to use fold changes—ie a 100uM BPA caused a 2-fold decrease in percent total motility from control. Also, percent or fold change may be more appropriate in the discussion and use mean and statistical values in the results.

Terms not typically used to describe changes:

significant lowering effect

significant reducing effect

reducing effect and softer effect

statistically significant—just use “significant”

Also, in results remove editorial words such as interestingly.

Example Line 89-

However, 100 µM BPS or BPA causes a significant lowering effect on total sperm motility at any time studied.  

Change to

However, 100 µM BPS and BPA significantly decreased total sperm motility at both 3 hours and 20 hours of incubation.  Next sentences should contain the Mean, F-test and P-values of the compared variables.

Line 202-207

Line 207 describing figure 8—Stated that it is Not a significant change – so not “clearly visible”.  If the p-value was presented and close to a significant p-value then could be mentioned that this variable for this treatment tended to be higher but the difference was not significant. 

3)      Figure information

Line 229-234

Figure 9. Effects of BPS and BPA in pig sperm plasma membrane lipid organization. Pig spermatozoa were incubated in TBM at 38.5 °C for 3 and 20 hours, with 5 % CO2 in the absence (control) or presence of BPA (1 and 100 μM) or BPS (1 and 100 μM). This experiment was performed 8 times (n = 8). The results are expressed in box-whiskers plots which shows the distribution of data into quartiles for the mean of relative fluorescence intensity (RFI) of M540 positive/Yo-Pro-1 negative spermatozoa. Boxes with different letters are statistically different from each other, so that for P < 0.05.

The words in red don’t describe the picture/graph correctly.—This is Fixed for Figure 10 and the rest of the figures. 

Author Response

Response: Both bisphenol concentrations were chosen based on different studies previously performed in this subject [4,31,38] and also on our own results obtained after performing a 3 h dose-effect curve of BPA, ranging from 10-12 M to 10-4 M, in porcine spermatozoa and analyzing BPA effect in sperm motility and kinetic p.arameters. Data showed that BPA inhibited between 10-6 y 10-4 M and that lower concentrations than 10-6 M did not significantly affect sperm motility. This is the reason why we have decided to used BPA concentrations 1 and 100 µM. We have added this information in the Method Section (line 482).

Response: We agree with the referee about this comment and basically that was our intention. The objective evaluation of sperm motility and its parameters has been used in numerous studies as a correct approach to study the functionality of pig spermatozoa. The parameters investigated (motility, ROS, ΔΨm, sperm plasma membrane lipid organization, protein phosphorylation) cover wide-range sperm features than allow a proper study of pig sperm function and thus, the most objective evaluation of the bisphenols in these gametes. Data are represented in box-whiskers plots because it allows the quick visualization of the data distribution into quartiles and the extreme values including also the median and atypical values of the data.

Grammar

-Example Line 55-57

Response: Corrections have been performed, as suggested.

-Line 57 I suggest changing “Moreover” to “However”

Response: The suggestion has been included.

-Line 55-68 remove the repetitive transition/connectors words – recently—author information may be best put in the cite reference.

Response: As suggested we have removed the connector.

-Line 70 “or” should be “and”

Response: The modification has been made.

Response: The values of F-test and P-values of the compared variables have been included in the Figure legends. 
Modifications suggested by the referee have been made, specifying whether the percentage is referred to values obtained in control or using fold-decrease when appropriate.

Response: As requested, modifications have been made in the text to correct these terms.

Response: We agree with the referee; the description of Figure 8 has been corrected.

Response: The referee is right as there was an error in mentioned Figures legend. This error has been corrected by including the proper legend in Figure 9 and we have revised the legend of the rest of figures.